# Monitoring the Calibration of In-Office 3D Printers

**DOI:** 10.3390/dj11010020

**Published:** 2023-01-05

**Authors:** Esha Mukherjee, Luke Malone, Edward Tackett, Bakeerathan Gunaratnam, Gerald Thomas Grant

**Affiliations:** 1Advanced Graduate Education Program Prosthodontics, University of Louisville School of Dentistry (ULSD), Louisville, KY 40202, USA; 2Additive Manufacturing Institute of Science & Technology (AMIST), University of Louisville J.B Speed School of Engineering, Louisville, KY 40292, USA; 3Bioinformatics and Biostatistics Department, University of Louisville School of Public, Health and Information Sciences, Louisville, KY 40202, USA

**Keywords:** stereolithography (SLA), desktop 3D printers, dental resins, print accuracy, calibration

## Abstract

Most desktop 3D printers lack features that allow manual calibration of printer parameters. It is crucial to assess the accuracy of printing to minimize the margin of error and variance between each print. Therefore, this study aimed to develop a method for monitoring the calibration of in-office 3D printers. A calibration coupon was designed to have a tolerance and dimensions that define nominal geometry and allow the measurement of variances occurring in X–Y axes and curvature. Ten printing cycles were run on two stereolithography (SLA) 3D printers with two different resins. Additionally, the coupons were positioned in five positions on the build platform to assess errors caused by differences in positioning. Measurements were made on the X and Y axes. No statistical difference was noted between the coupons being printed in different positions on the build platform and between the two resins at both X and Y axes of measurement (p > 0.05). Desktop 3D printers currently lack a standardized calibration protocol, which provides a closed loop for design and manufacturing of printed parts. The coupon in this study will allow monitoring the calibration of desktop 3D printers to ensure high-quality printing.

## 1. Introduction

Three-dimensional (3D) printing has become increasingly popular over the years due to reduced print costs and a more efficient print rate [1,2]. In 1983, Charles Hull printed the first three-dimensional object using a stereolithography printer [3]. This received increased attention in the field of architecture and aeronautics [4], with the potential for the construction of parts that needed millimetric precision. It started gaining popularity in the field of medicine in the 1990s [5]. Today, it has several applications in the field of dental medicine due to advancements in computerized scanning. Digital data from intraoral/desktop scanners and cone beam computerized tomography (CBCT) is used to design and manufacture prosthetic components, thereby essentially reducing the need to outsource laboratory work to dental technicians [6].

Stereolithography (SLA) was the first commercial printer available for rapid prototyping [7]. The apparatus consists of a scanning laser that builds parts one layer at a time in a tank of photopolymerizable resin. Each layer is traced out by the laser on the liquid resin, after which the build platform descends into the tank, and a new layer of resin is wiped over the surface. This process repeats itself until all the layers are complete [8]. It is necessary to generate supports on the build software to allow resistance against gravity as well as the action of wiping. Postprocessing steps include washing the parts with alcohol to remove the excess resin, followed by curing any uncured resin in a UV oven. This technology is predominantly used in the fabrication of implant surgical guides [1,9].

Today, there are several other types of printers that use different mechanisms for printing. Selective laser sintering (SLS), fused deposition modeling (FDM), photopolymer jetting (PPJ), and digital light processing (DLP) are some of the technologies that are available on the market today and have specific applications based on the material to be printed as well as the desired speed and accuracy [1,10,11] (Table 1).

There are several applications of 3D printing in dentistry. These include, but are not limited to fabrication of dental models, drilling and cutting guides for implant dentistry, crown copings and partial denture frameworks, interim and definitive removable prostheses, provisional crowns, and aligners for orthodontic treatment [12,13]. Low manufacturing costs and less waste of materials used are some of the enticing factors that have led to its popularity compared to additive manufacturing in dentistry [14].

Some applications require printed parts to have higher accuracy than others, for example, definite models of anatomic structures and surgical guides. This necessitates high reproducibility and contouring accuracy [6,15]. There has been a considerable shift toward the usage of in-house 3D-printed surgical guides compared with laboratory-fabricated guides [16]. This reduces the overall cost as well as turnover time. These guides have shown similar accuracy with implant placement when compared with laboratory-fabricated guides. Industrial-grade 3D printers can be calibrated and verified after each print using a quality control system that provides a closed loop for the design and fabrication process. Most in-office printers for dental use lack features that allow adjustment of individual parameters to ensure accurate and precise printing each time. This suggests that inaccuracies may often go unnoticed, which in turn would affect the quality of treatment we render to our patients [17].

The accuracy of 3D printers is affected by several factors such as the material used, geometric features and topology of the object, nominal dimensions, wall thickness, solid/shell, and postprocessing techniques. There have been several studies designed for the development of “test cubes” [6] or benchmark parts [18] for the performance evaluation of 3D printers. These parts essentially help with examining the accuracy and making necessary adjustments to the printing parameters of the system [6,19]. However, most of the designs described by these advanced journals are complex. They may not be convenient for someone who lacks the appropriate training and knowledge in rapid prototyping. This is true for most dentists as well as technicians. Hence, it is imperative to develop a standard quality control protocol that is simple and easy to use by dental professionals [20].

The study aimed to address this issue by developing a calibration coupon to assess the effect of different resins used for printing and the position of the test object on the build platform on printing accuracy. Variances were assessed for coupons printed in two different resin materials. As printing may vary based on the position of the test object on the build platform, the variances were also measured for coupons printed on five different positions on the platform [5]. Null hypotheses are that variances are equal at all five positions on the build platform at the X and Y axes for both Dental SG and Grey V4 resins and that they are equal between the Grey V4 and Dental SG resins at the first position at X and Y axes.

## 2. Materials and Methods

The following materials, equipment, and software were used for the study:

### 2.1. Materials

Slicing software (Chitubox, Shenzhen, China) for designing;2 SLA 3D printers (Form 2, Formlabs, Somerville, MA, USA);Formlabs Dental SG and Grey V4 resins (Formlabs, Somerville, MA, USA);Resin cleaning station (Veri Wash, Whip Mix, Louisville, KY, USA);99% isopropyl alcohol(Florida laboratories inc., Fort Lauderdale, FL, USA)Digital calipers. (IGaging Absolute Origin, IGaging, Los Angeles, CA, USA)

### 2.2. Designing the Coupon

It was decided to keep the design simple for measurements and small enough to be printed with each printing cycle to monitor calibration. This is important because the coupon should not occupy a significant amount of space on the build platform and, at the same time, should not be too fragile and susceptible to distortion. Once the initial design was approved, slicing software (Chitubox, Shenzhen, China) was used to design a calibration coupon to help measure X–Y axes and curvature variances. However, curvature variances were not assessed in this study (Figure 1). Z-axis is considered controlled by the mechanical components of the build platform and is, therefore, not an issue of calibration [21].

The final size was determined based on the x–y resolution of the printer as well as the ease of measurement. Several trial printing cycles were carried out with increasing dimensions. The starting dimension was 12 millimeters (mm) × 12 millimeters (mm) × 1 millimeter (mm). However, this was too fragile to carry out the measurements. Based on test runs, the coupon size was set as 24 mm × 24 mm × 2 mm (Figure 1). This was then subjected to further testing, as mentioned below.

To test the printability of the calibration coupon, printing cycles were carried out on an LCD-based SLA 3D printer (Anycubic Photon, Anycubic technology co, Hongkong) at Additive Manufacturing Institute of Science and Technology (AMIST). The coupon was positioned on four corners of the build platform and the center. Five printing cycles were run, and the printed coupons were post-processed based on the manufacturer’s recommendations. One thing that was noticed was that printing the coupons directly on the platform resulted in difficulty taking the coupon off the platform after the printing cycle was completed and often led to chipping. Hence, it was decided to add mini rafts on the coupons to allow for easy removal of supports and also to minimize any inherent defects before measurements (Figure 2). Followed by this, another set of test printing cycles was carried out on an SLA 3D printer (Form 2, Formlabs, Somerville, MA, USA) printers at the University of Louisville School of Dentistry (ULSD). This was conducted because the final experiment was to be performed on this specific 3D printer. There were inconsistencies noted in the printed coupons. In four out of the five test runs, the coupon in the center failed to print, and the ones at the four corners were distorted (Figure 3). The particular SLA 3D printer (Form 2, Formlabs, Somerville, MA, USA) has a “heatmap” feature on the account dashboard, which allows the provider to visualize different areas of the resin tank and their laser exposure level. If a particular position is used consistently for printing, it will appear darker on the map, which represents more print layers for the parts printed compared with the other positions. This reduces the translucency of the tank and affects the quality of the print. Using the heatmap feature, it was observed that the center of the tank was used more frequently than the corners, which reduced the tank’s translucency over time (Figure 4). Another possible explanation could be partial curing of the uncured resin in the tank. Additionally, the resin cartridge had not been changed for more than 3 months, which is more than the recommended shelf life. Once the resin tank was replaced and a new resin cartridge was inserted, there were no more noticeable distortions. The final dimensions of the coupon were set as 24 mm × 24 mm × 2 mm. The inner diameter was 4 mm, and the outer diameter was 16 mm (Figure 1).

To help with the orientation of the coupon for measurement, design software (Meshmixer software, Autodesk, San Rafael, CA) was used for engraving purposes. The letters “X” and “Y” were engraved on two of the vertical extensions/strut portions. Numbers “1” to “4” were engraved on the circular part of the coupon. The numbers allowed for orientation after taking the coupons off the platform and enabled performing measurements in defined areas of the coupon (Figure 5).

### 2.3. Positioning of Coupon and Printing Cycles

Two SLA 3D printers (Form 2, Formlabs, Somerville, MA, USA) were loaded with two commonly used dental resins used for fabricating surgical guides (Dental SG, Formlabs, Somerville, MA, USA) and dental models (Grey V4, Formlabs, Somerville, MA).

To evaluate whether there were any changes in print accuracy on changing the position of the coupon across the build platform, the coupon was strategically placed on the four corners and the center of the build platform. The SLA 3D printer (Form 2, Formlabs, Somerville, MA, USA) has a 405 nm laser that is mounted on one corner of the printer. The objective of positioning the coupons in four corners and the center was to evaluate any changes in print accuracy with the distance from the laser source. Increased distance could cause a reduction in laser intensity as well as laser beam refraction. The coupon was positioned horizontally with mini rafts as support (Figure 2). The print resolution was set to 50 micrometers for both resins. Ten printing cycles were completed consecutively on each printer. There were five coupons per printing cycle. This resulted in a sample size of 50 coupons per resin group and a total sample size of 100 coupons. After each print, the coupons were washed in 99% isopropyl alcohol for 10 minutes for dental model resin (Grey V4 Resin, Formlabs, Somerville, MA, USA) and 20 minutes for surgical guide resin (Dental SG Resin, Formlabs, Somerville, MA, USA). The coupons were allowed to dry, and supports were removed (Figure 6 and Figure 7).

### 2.4. Measurement of the Coupon

Measurements were made along the x- and y-axis with digital calipers up to two decimal points (Table 2). The lower jaws of the caliper were oriented between two vertical struts of the coupon, i.e., for measurement on the x-axis, one jaw extended to the vertical strut engraved with the letter “X.” The other jaw extended to the opposing strut. Similarly, for the y-axis measurement, the lower jaws of the caliper extended between one strut marked “Y” to the opposite strut (Figure 8).

### 2.5. Statistical Analyses

Once the information was collected for 10 cycles, the measurements of positions for both resins at the X and Y axes were entered into SPSS version 28.0. Instead of comparing the variance between Grey V4 and Dental SG, we compared the variance for positions 1 through 5 on the build platform for both resins at the X and Y axes, i.e., the coupon printed in one resin (Dental SG, Formlabs, Somerville, MA, USA) at X-axis position 1 was compared with the coupon printed in another resin (Grey V4, Formlabs, Somerville, MA, USA) at X-axis at the same position. The analysis was completed for all positions at the x-axis in the same manner. Once that was achieved, corresponding position measurements were compared at the y-axis between both resins.

In order to compare the variance between the two resins (Dental SG and Grey V4, Formlabs, Somerville, MA, USA) at the X and Y axes, Levene’s test for equality of variances was used at a 5% level of significance.

## 3. Results

Table 2 shows the X–Y axes measurements made for coupons printed in both the resins at five positions on the build platform. It can be noticed that the measurements were closer to the original coupon dimensions (24 × 24 mm) for one of the resins (Grey V4, Formlabs, Somerville, MA, USA) compared with the other (Dental SG, Formlabs, Somerville, MA, USA).

The mean values for the 10 printing cycles for coupons printed in Grey V4 Resin (Formlabs, Somerville, MA, USA) at the x-axis from positions “1” through “5” ranged from 23.81–23.82 mm. Y-axis measurement mean values ranged from 23.80–23.85 mm (Table 3).

X-axis measurements at positions 1 through 5 for Dental SG Resin (Formlabs, Somerville, MA, USA) ranged from 23.63–23.67 mm (mean value). For the y-axis, the mean value ranged from 23.62–23.69 mm (Table 3).

### 3.1. Comparison of Two Different Resin Materials

Table 3 shows the descriptive statistics of all positions at the X and Y axes. Since the *p*-value for Levene’s test is greater than the significance level (0.05), we can conclude that there is no statistically significant variance difference between the coupons printed in Dental SG and Grey V4 resins at the X and Y axes (*p* > 0.05). the X and Y axes.

### 3.2. Comparison of Different Positions on the Build Platform

Since the *p*-value for Levene’s test is greater than the significance level (0.05), we can conclude that there is no statistically significant variance difference between the printed coupons at positions 1 through 5 on the build platform for both Dental SG and Grey V4 resins at the X and Y axes (Table 4 and Table 5) (*p* > 0.05).

## 4. Discussion

There is a trend toward greater use of digital technology in dentistry, with a gradual shift from subtractive manufacturing to additive manufacturing [22]. This has led to an increased number of desktop 3D printers on the market, and monitoring performance and quality have become equally critical. Most companies use the printer’s resolution as a luring factor to sell their product. However, having a higher resolution in the printer specification does not guarantee accuracy. It is imperative to understand the terms accuracy, which comprises precision and trueness [23], and the tolerance of the printer. Precision refers to “the variability between repeated measurements” [23,24]. Trueness is defined as “the closeness of agreement between the average value obtained from a large series of test results, and an accepted reference value” [23]. On the other hand, tolerance is defined by an acceptable variance in the printer’s precision, which depends on the specific application [25]. For example, the tolerance range for printing models can be slightly more than the tolerance range for surgical guides, requiring higher accuracy for acceptable clinical results. This allows us to understand possible errors within the 3D printing process. The printer may be performing well within the tolerance range of a project, but it may not necessarily be precise. Similarly, sometimes there may be precision in the process, but the parts may not be true to the original or reference file. Some of these errors may be due to factors such as the choice of the 3D printer technology, the material used, and postprocessing. However, it could also be due to an inherent issue with the calibration of the 3D printer [24].

Evaluating accuracy has become necessary with the increased use of 3D printing technology to fabricate surgical guides. When comparing different additive technologies for fabricating surgical guides, namely PolyJet, SLA, and digital light processing (DLP), PolyJet had the best outcomes in terms of 3D deviations at the entry point and the apex. The result was statistically significant when comparing PolyJet and DLP but not between PolyJet and SLA. DLP printers, on the other hand, had the fastest processing time. However, other studies have shown no significant difference between different technologies [26,27,28].

In this study, a calibration coupon of specific dimensions was designed that allows the measurement of variances on X/Y axes. The tolerance was set as ±0.1mm, which was described as clinically acceptable [29]. Additionally, the coupon was placed in five different positions on the build platform, and pre-cure measurements were made to assess any differences in variances with the positioning of the coupon at varying distances from the mounted laser of the 3D printer.

Results showed no significant difference with the coupon placement, and there were no significant differences between the two types of resin materials that were tested. However, when individual measurements were compared between all 10 cycles, it was observed that the coupons printed with Dental SG Resin (Formlabs, Somerville, MA, USA) were less true to the original dimension of the coupon (up to 0.5 mm discrepancy) compared with the Grey V4 Resin (Formlabs, Somerville, MA, USA), which had up to 0.26 mm discrepancy (Table 2). Additionally, coupons printed in both these materials lacked precision. This suggests that material properties may affect the accuracy and overall print quality; however, it was not statistically significant in this study. Materials with lower elastic modulus tend to have more dimension change before curing. This resin was discontinued by the manufacturer recently and was replaced with another resin (Surgical Guide Resin, Formlabs, Somerville, MA, USA) with a comparatively higher elastic modulus (>2400 MPa for Surgical Guide Resin vs. ~1500 MPa for Dental SG Resin) [30]. When compared with Grey V4 Resin, which has a flexural modulus of 2.2 GPa [31], Dental SG Resin has a lower modulus and is more susceptible to deformation before postcuring. Furthermore, the distance of the coupon from the mounted laser did not result in any significant differences in accuracy. This potentially allows the placement of the calibration coupon in any of the five positions for assessing accuracy. However, this needs to be verified with an increased sample size. Enough evidence was not present to reject the null hypotheses.

The coupon itself was deemed helpful in understanding the overall performance of the 3D printer. An interesting aspect was noticed during the initial test printing cycle of the coupon on the SLA 3D printer that had an old resin tank, which had exceeded its recommended use. The printed coupons were visually deformed, and some of them even failed to print after multiple attempts. This suggests the importance of following the manufacturer’s recommendations for maintenance of the printer and its parts, resin tank, and timely change of resin cartridge. It may be suggested to position the parts in different areas on the platform on SLA 3D printers to extend the tank’s lifetime and improve print quality. This, along with following the correct postprocessing instructions mentioned for each material, significantly affects the accuracy and strength of the object printed. Following these set recommendations usually takes care of noticeable errors in 3D printing.

In the dental literature, studies on the calibration of 3D printers are lacking. However, it is undoubtedly the need of the hour, with a noticeable shift of interest toward additive manufacturing applications in dentistry today. Most dental staff and clinicians rely on the “autocalibration” feature of 3D printers; however, it is imperative to have a calibration protocol in place to ensure that inaccuracies or inconsistencies in the printing process are identified and tackled in an effective and timely manner. The coupon in this study may be used to assess the calibration and overall performance of the 3D printer. The printer’s performance can be optimized first by following the manufacturer’s recommended maintenance protocols. If issues related to accuracy are noticed despite a strict maintenance workflow, support staff may be contacted to assist with the following steps, which may potentially involve recalibrating the printer. These steps toward monitoring the calibration of the 3D printers eventually help render high-quality treatment to our patients.

Drawbacks of the study include a relatively small sample size. Additionally, recording the measurements with a digital caliper is subject to human error. Since this study only focused on the X and Y axes, studies need to be designed for measuring variances in curvatures. Measuring curvature variance is essential for most objects printed for dental use, for example, the guide sleeve holes for implant surgical guides.

In future studies, machined quick-check GO NO-GO calibration gauges with a specific tolerance range can be manufactured. This is an inspection tool commonly used in the engineering field. It allows for quick checks of parts to assess whether they fall within the desired tolerance range. It involves two tests. The printed part needs to pass one test (GO) and fail the other test (NO-GO). For example, the coupon set at a tolerance range of +/−0.1 mm for X–Y axes measurements will have a gauge with a “GO” end at 24.1 mm opening and a “NO-GO” end with 23.89 mm (slightly less than the lower limit of tolerance range) (Figure 9). The coupon that fits the “GO” end of the gauge and does not fit the “NO-GO” end falls within the tolerance range of 0.1 mm and passes the test. The same can be performed for the measurement of curvature variances. These tools will simplify the checking process for dental staff and could be reliably used in comparison to digital calipers. Additionally, the current study may be extended to study the post-cure changes of the coupons to understand potential issues of shrinkage associated with the curing process. In addition, a longer duration between printing cycles would allow for studying any changes that may occur to the resin once it is already dispensed into the resin tank. There is a scope for expansion in several directions, as mentioned above.

## 5. Conclusions

Several factors dictate the success of 3D printing. Since most in-office 3D printers do not have a manual calibration feature, it leaves us with very little control. However, some manufacturers’ recommendations are essential to ensure high-quality printing. This includes but is not restricted to the timely change of the resin tank and resin cartridge, following postprocessing instructions specific to each material, and checking the heat map, if available, for strategic positioning of the parts to be printed to avoid overuse of certain areas. Overusing the tank can alter the quality, as was observed in this study.

Quality control is crucial in ensuring accurate printing, and it is currently lacking with medical- and dental-grade printers. The coupon designed in this study will allow the provider to assess the accuracy of the print. This study showed no significant difference in coupon placement on the platform. Hence, it may be placed on any of the five positions mentioned in the study along with the object to be printed, or it may be printed alone. If the coupon falls outside the tolerance range, first, it must be assessed whether the manufacturer’s recommendations were followed for the printing cycle as well as postprocessing. If there were any shortcomings in the protocol, the printing cycle should be repeated after making the necessary changes. If the problem persists, the printer’s manufacturer may be contacted to resolve any technical issues that may hamper the accuracy or to recalibrate the printer altogether. In this study, the potential errors that led to the loss of trueness and precision of the printed coupons with one of the resins (Dental SG, Formlabs, Somerville, MA, USA) could be attributed to the overall properties of the material, especially the flexural modulus. Hence, material selection based on the specific needs of the project plays a key role in the overall success of the 3D printing process. Having a standardized calibration protocol for in-office 3D printers will allow clinicians to provide quality treatment to their patients. Further studies are needed to evaluate the application of this coupon with different types of 3D printers and a broader spectrum of resin materials.

## Figures and Tables

**Figure 1 dentistry-11-00020-f001:**
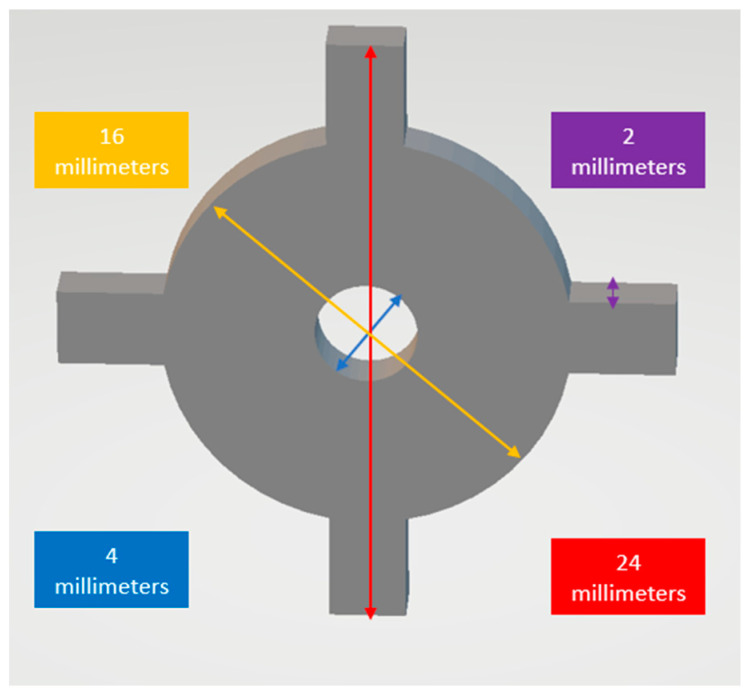
Final digital design of the calibration coupon showing the overall dimensions.

**Figure 2 dentistry-11-00020-f002:**
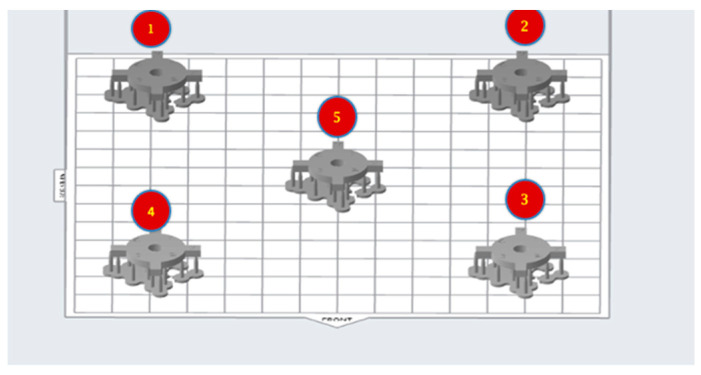
The layout of the designed coupons on the build platform of the 3D printing software (PreForm software, Formlabs, Somerville, MA, USA). Numbers were added using Microsoft PowerPoint software (Microsoft Corporation, Redmond, WA, USA).

**Figure 3 dentistry-11-00020-f003:**
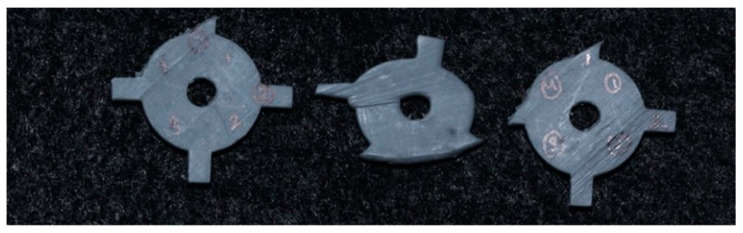
Distorted calibration coupons during the experimental study on the SLA 3D printer. The coupon on the far left shows incomplete printing of the top vertical strut. The middle coupon shows distortion in overall shape and is missing the struts on the right side and on the bottom. The coupon on the far right shows incomplete printing of the vertical struts on the left side and on the top.

**Figure 4 dentistry-11-00020-f004:**
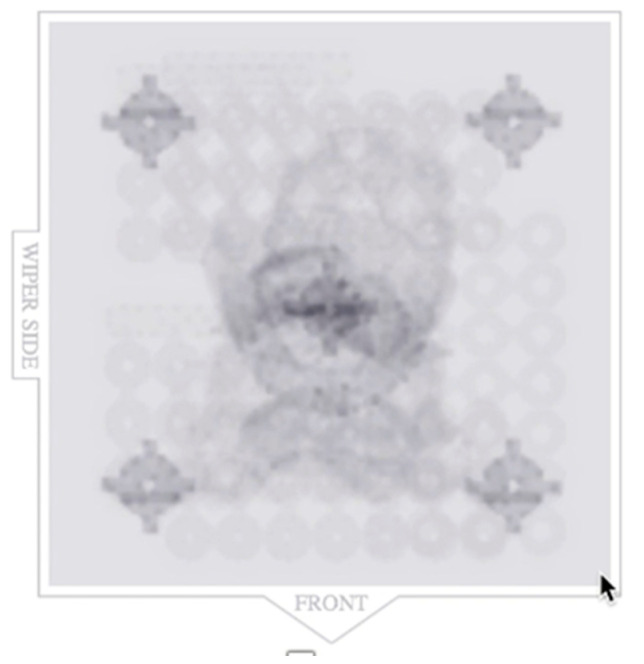
Heat map feature on the dashboard of 3D printing account (Formlabs, Somerville, MA, USA) showing placement of the previously printed objects on the build platform.

**Figure 5 dentistry-11-00020-f005:**
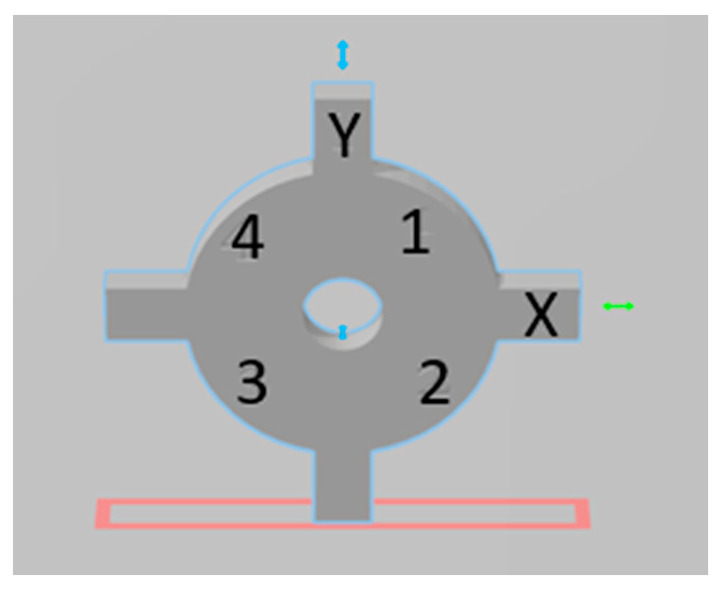
The final design of the calibration coupon with letters “X” and “Y” and numbers “1” through “4” engraved on the coupon (Meshmixer software, Autodesk, San Rafael, CA) to allow for orientation and measurement of the coupons.

**Figure 6 dentistry-11-00020-f006:**
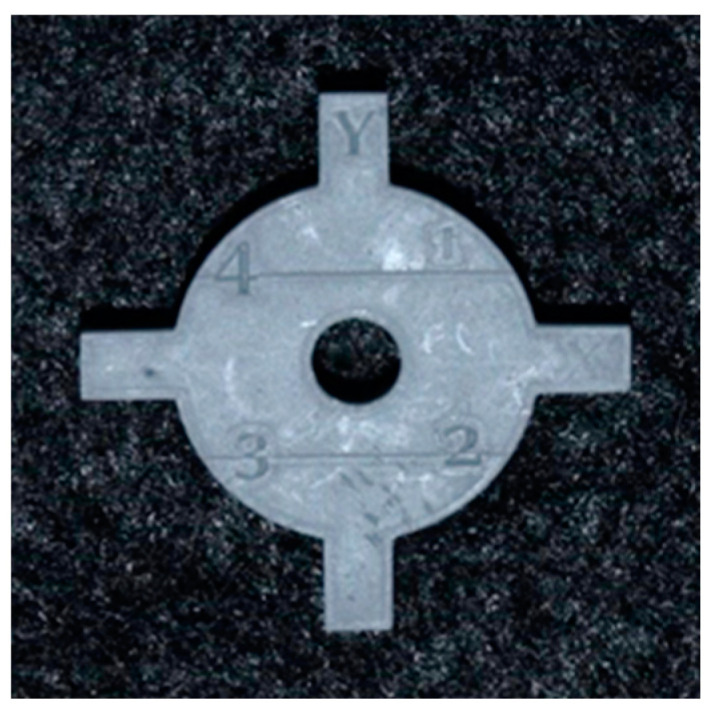
Final printed coupon in the resin for dental models (Grey V4 Resin, Formlabs, Somerville, MA, USA).

**Figure 7 dentistry-11-00020-f007:**
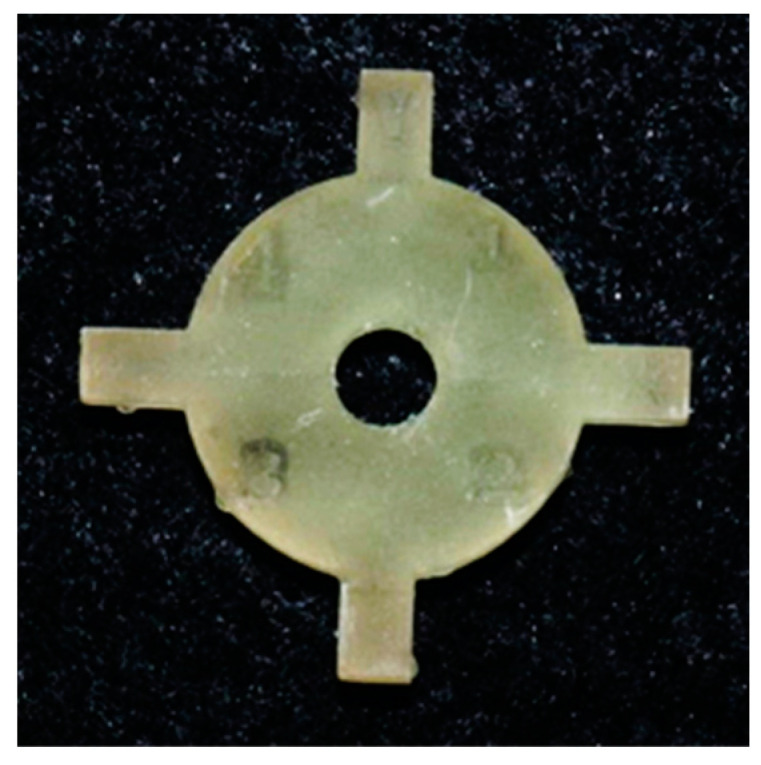
Final printed coupon in the resin for surgical guides (Dental SG Resin, Formlabs, Somerville, MA, USA).

**Figure 8 dentistry-11-00020-f008:**
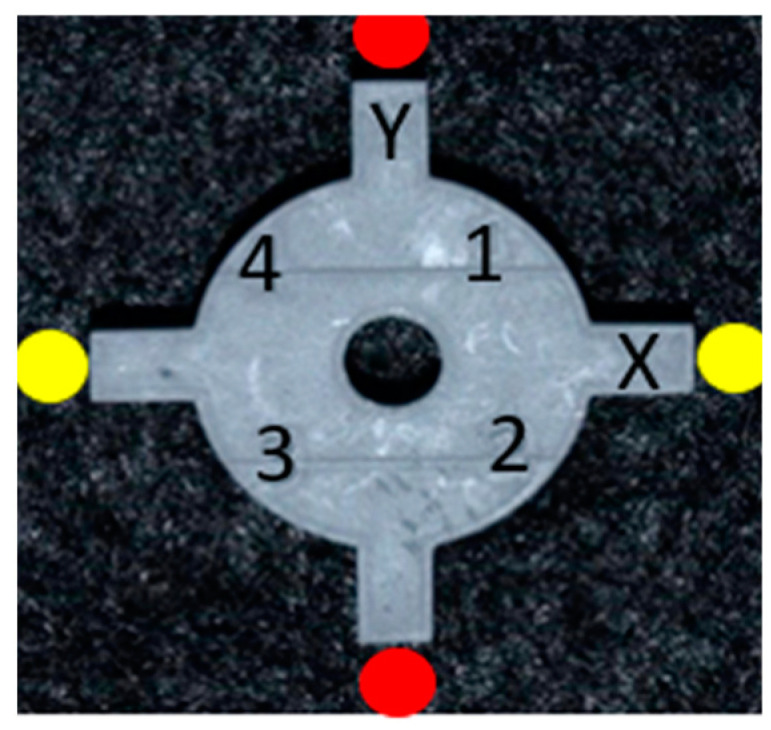
Picture denoting the orientation of digital caliper jaws for measurement. The red dots show the orientation of the lower jaws of the calipers to measure along the Y-axis. The yellow dots show the orientation of the lower jaws of the calipers to measure along the X-axis.

**Figure 9 dentistry-11-00020-f009:**
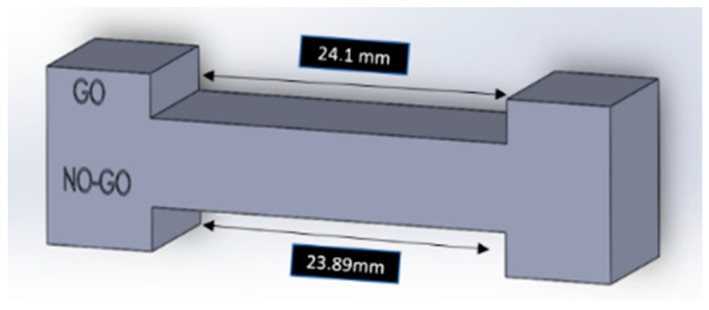
GO-NO calibration gauge designed for the X–Y measurement of coupon. The GO end has an opening of 24.1 mm, and the NO-GO end has a 23.89 mm opening.

**Table 1 dentistry-11-00020-t001:** Summary of 3D printing technologies.

3D Printing Technology	Mechanism	Advantages	Limitations
Stereolithography (SLA)	Liquid photopolymer in a vat is selectively cured by light-activated polymerization	Low cost, good surface quality of the print, and high resolution	Need for extensive postprocessing, longer print times
Selective laser sintering (SLS)	Thermal energy used for selective fusion of regions on a powder bed	Excellent surface quality, large build volumes, no supports, minimum postprocessing	High equipment cost, challenging to operate, maintain, and calibrate
Fused deposition modeling (FDM)	Selective extrusion of melted material through a nozzle or ban orifice	Easy to operate, wide spectrum of thermoplastic materials can be printed, used for bioprinting	Long print times, relatively lower print resolution
Photopolymer jetting (PPJ)	Layers of photopolymer laid down and light cured with every passage of printer head	High resolution, large build volume, wide spectrum of materials can be printed, multicolor printing, multiple print heads allow printing of complex structures	High cost of equipment and maintenance, printed objects are brittle in nature
Digital light processing (DLP)	Similar to SLA	Good surface finish, high accuracy, faster than SLA	Need for postprocessing

**Table 2 dentistry-11-00020-t002:** X–Y axes measurements of the coupons printed in two resins (Dental SG and Grey V4, Formlabs, Somerville, MA, USA) at five different positions on the platform.

Printing Cycle	Position on the Platform	Dental SG Resin		Grey V4 Resin	
		X-Axis (In Millimeters)	Y-Axis (In Millimeters)	X-Axis (In Millimeters)	Y-Axis (In Millimeters)
1	1	23.64	23.68	23.82	23.84
	2	23.66	23.61	23.8	23.81
	3	23.72	23.69	23.79	23.78
	4	23.69	23.7	23.81	23.77
	5	23.67	23.65	23.81	23.82
2	1	23.63	23.62	23.81	23.79
	2	23.63	23.63	23.81	23.76
	3	23.66	23.72	23.78	23.78
	4	23.64	23.76	23.77	23.72
	5	23.62	23.64	23.78	23.82
3	1	23.5	23.53	23.82	23.77
	2	23.54	23.57	23.81	23.84
	3	23.59	23.68	23.77	23.77
	4	23.6	23.62	23.79	23.79
	5	23.63	23.57	23.85	23.84
4	1	23.66	23.65	23.76	23.77
	2	23.65	23.65	23.77	23.75
	3	23.65	23.69	23.75	23.74
	4	23.66	23.7	23.74	23.77
	5	23.64	23.56	23.78	23.73
5	1	23.54	23.58	23.8	23.83
	2	23.5	23.61	23.85	23.8
	3	23.6	23.71	23.82	23.79
	4	23.61	23.59	23.79	23.84
	5	23.53	23.57	23.83	23.84
6	1	23.72	23.71	23.86	23.87
	2	23.77	23.7	23.87	23.88
	3	23.74	23.84	23.83	23.85
	4	23.7	23.78	23.81	23.84
	5	23.71	23.69	23.85	23.9
7	1	23.67	23.63	23.84	23.84
	2	23.68	23.68	23.81	23.88
	3	23.68	23.77	23.81	23.82
	4	23.67	23.73	23.86	23.81
	5	23.69	23.66	23.9	23.9
8	1	23.61	23.64	23.86	23.84
	2	23.64	23.64	23.85	23.86
	3	23.66	23.74	23.85	23.84
	4	23.63	23.73	23.83	23.84
	5	23.6	23.59	23.84	23.88
9	1	23.65	23.65	23.87	23.87
	2	23.67	23.68	23.84	23.9
	3	23.68	23.78	23.86	23.83
	4	23.68	23.74	23.86	23.86
	5	23.68	23.67	23.84	23.9
10	1	23.65	23.69	23.85	23.78
	2	23.68	23.67	23.84	23.81
	3	23.72	23.75	23.75	23.73
	4	23.66	23.7	23.8	23.77
	5	23.68	23.65	23.79	23.81

**Table 3 dentistry-11-00020-t003:** Descriptive statistics of positions for Dental SG and Grey V4 resins at.

Resin Material	Dimension	Position	N	Min	Max	Mean	Median	SD
Grey V4	X	1	10	23.76	23.87	23.83	23.83	0.034
		2	10	23.77	23.87	23.83	23.83	0.032
		3	10	23.75	23.86	23.81	23.81	0.039
		4	10	23.74	23.86	23.81	23.81	0.039
		5	10	23.78	23.90	23.83	23.84	0.036
	Y	1	10	23.77	23.87	23.83	23.84	0.032
		2	10	23.75	23.90	23.83	23.84	0.051
		3	10	23.74	23.85	23.80	23.80	0.035
		4	10	23.72	23.86	23.80	23.81	0.043
		5	10	23.73	23.90	23.85	23.81	0.043
Dental SG	X	1	10	23.50	23.72	23.63	23.64	0.064
		2	10	23.50	23.77	23.64	23.66	0.075
		3	10	23.74	23.59	23.67	23.67	0.049
		4	10	23.60	23.70	23.65	23.66	0.033
		5	10	23.53	23.71	23.64	23.66	0.055
	Y	1	10	23.53	23.71	23.64	23.64	0.053
		2	10	23.57	23.70	23.64	23.64	0.040
		3	10	23.68	23.84	23.74	23.73	0.050
		4	10	23.59	23.78	23.69	23.70	0.061
		5	10	23.56	23.69	23.62	23.62	0.048

**Table 4 dentistry-11-00020-t004:** Significance of Levene’s test when comparing different positions on the build platform at the X-axis according to the *p*-value.

Position on the Build Platform	*p*-Value
1	0.197
2	0.167
3	0.592
4	0.097
5	0.167

**Table 5 dentistry-11-00020-t005:** Significance of Levene’s test when comparing different positions on the build platform at the Y-axis according to the *p*-value.

Position on the Build Platform	*p*-Value
1	0.289
2	0.348
3	0.348
4	0.278
5	0.732

## Data Availability

The data presented in this study are available on request from the corresponding author. The data are not publicly available.

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
