# Peer review of "Monitoring the Calibration of In-Office 3D Printers"

_dentistry, 2023, doi:10.3390/dj11010020_

Round 1
Reviewer 1 Report
This manuscript is well-organized. But, in my thought, the number of samples in this study is so limited. (as you mentioned 'ten cycles' )
I recommend to increase number of samples for statistical significane. Or you have to describe the number of samples.
1. Please describe why you set that coupon size specifically and correct figure 1 (24m -24mm). 2. The authors have to explain the number of samples. 3. Please describe the statistical method you used in this study. 4. Please explain whether there is a special reason for setting the coupon location to 5 places.
Author Response
Thank you for taking the time to review our paper. We appreciate your valuable feedback. We have considered your comments and made the necessary changes to the manuscript.
- Please describe why you set that coupon size specifically and correct figure 1 (24m -24mm).
The final size was determined based on the x-y resolution of the printer as well as the ease of measurement. Several trial printing cycles were carried out with increasing dimensions. The starting dimension was 12 millimeters(mm) x 12 millimeters(mm) x 1 millimeter(mm). However, this was too fragile to carry out the measurements. Based on test runs, the coupon size was set as 24mm × 24 mm × 2mm. This was then subjected to further testing.
- The authors have to explain the number of samples.
Ten printing cycles were completed consecutively on each printer. There were five coupons per printing cycle. This resulted in a sample size of fifty coupons per resin group and a total sample size of 100 coupons
- Please describe the statistical method you used in this study.
The descriptive analysis has been added to the manuscript and is marked in red.
- Please explain whether there is a special reason for setting the coupon location to 5 places.
The SLA 3D printer (Form 2, Formlabs; Somerville, MA) has a 405 nm laser that is mounted on one corner of the printer. The objective of positioning the coupons in four corners and the center was to evaluate any changes in print accuracy with the distance from the laser source. Increased distance could cause a reduction of laser intensity as well as laser beam refraction.

Reviewer 2 Report
Dear colleagues,
thank you for the opportunity to read your interesting manuscript about the calibration of In-Office 3D printers using calibration coupons. The presented work has scientific significance for the field, but needs major revision to be accepted for publication. Please revise your manuscript according to the enclosed comments:
1.) Title: As far as I know, "Geometric Dimensioning and Tolerancing" is a standardized system defined by the American Society of Mechanical Engineers, which does not only include metric measurements for defining the dimensional requirements for manufactured parts. GD&T describes nominal geometry and its allowable variation using a language or engineering drawings. The aim of GD&T is to add quality and reduce cost at the same time through producibility. It appears that the presented study uses a simpler way to assess the accuracy of printed objects and does not apply the rules stated in ASME Y14.5-2009. I think the methods presented have little connection to GD&T. Therefore, please consider revising the title.
In general: Please add mor appropriate references (up to 30 ca.)
2.) Abstract:
Line 22: Please specify that only two different resins were assessed.
3. Introduction:
Line 37: Please add references regardng the increasing attention in architecture, aeronautics and medicine.
Lines 56-79: As this study investigates SLA printers it seems a bit off-topic to introduce all of them with multiple sentences. Moreover, after description of each printing methods you should add an appropriate reference, which enables the reader to proof your statements and gather further infomation (Only at some points in this paragraph Refernce 1 is indicated). I would suggest to drastically shorten this paragraph at line 55 and create a short informative table with the characterisitcs of each of the printing methods. Please add appropriate references in line 51-55 after first mentioning of each method.
Line 80-85: Your statement is correct. Anyway in scientific literature you need to "proof" your statements by references. Please add current literature data.
Line 86-90: Please add reference regarding the shift to in-house printing.
Line 90-97: Very important and well written purpose of your study!
Line 103 ff. Please describe the term "worpiece with radii".
Line 110-120: This paragraph contains the aim of your study (imprtant and correct!), but also some M&M descriptions (line 112-115). Please remove any aspects of this paragraph, which describe how you've performed your investigation. This belongs to M&M. Focus on describing your aims shortly and concise, without further justification. At the end of your introduction no further explanations should be needed to understand the study aims.
3.) M&M
2.1: please add an intrducing sentence before listing your material. Please specify the type of the used printer (Form 2?).
2.2 This paragraph (design of the coupon) should include all aspects that affected the design of the final coupon. Please move any descriptions about your preliminary experiences (mini rafts, exceeded tank etc.) into paragraph 2.2 and close with the description of the final coupon design.
Line 137: Add explanation and reference indicating that Z-axis is "not as critical" as the other axes.
Figure 1: Consider Text editing. Captions should be at the same page as the figure. Please revise this figure. E.g.: mm (millimeter) and m (meter); spatial relations of the figure (a 6mm ring would appear much more slighter than presented, add the distances 16mm and 4mm --> where can they be measured?). It would be nice to have an appropriately captured screenshot of the meshmixer view instead of the "sketch" presented.
2.3: Consider deleting this paragraph after implementing all crucial aspects in 2.2!
Figure 2: Please specify what went wrong with each of the coupons. Upload higher resolution image please.
Figure 3: Upload higher resolution image please.
2.4
Lines 169-174: Consider splitting up this really long sentence. Btw do you mean "[...], as in corresponing studies."
Figure 4: Again, resolution is too low. It would be helpful and nice for visualization to have one proper view ot the coupon as displayed in your design software and adding all explanatory captures. Why did you add numbers 1-4?
Lines 179-191: It does not besome clear which measurements were performed indetail. What have you done with the caliper? What was documented --> Please report in results! Is there any explanation why you frequently speak about curvature measurements and warping but do not measure it? If possible it would be helpful to have a statement/ quantification about the warping of the parts! You do mention that warping is crucial, this obligates you somehow to quantify/ describe it. Can you add this asprect?
Line 189-191: Consider rewording or splitting up sentence.
Figure 6/7: Figures are of low quality. Could you take a photo in a light box? Please correct captions of figure 6 (typo).
Line 189: Fig. 6/6 seem not appropriate for visualizing the performed measurements. It would be helpful to have a figure describing the performed measurements.
4.) Results:
3.1 Unfortunately you do mention the importance of warpage for accuracy of 3D printing, but don't measure it. Apparently, the statistical analyses regarding x/y axes do not concern warpage. As already stated I would recommend to add measurements/ a statement about the quantity/extent of warpage. Mere visual examination is a comparable weak method.
Table 1: Please add the name of the actual analysis performed (Levene's test?) The p-value gives the level of significance but is not considered to be an an analysis itself. --> Same for Table 2 and 3. Could you please describe which measurements where compared with each other? Usually you set one measurement (e.g. position 1) as reference and compare it to the other. How did you perform the measurement? In general: please present the actual results of the measurement performed (using the caliper) in a concise table. This is a crucial drawback of your manuscript - it does not contain any numbers regarding the measurement, only statistical analysis. Please revise.
5.) Discussion
Lines 230-238: Consider using shorter sentences when introducing the terms trueness and precision. Maybe you can write it as it is usual in definition phrases. Consider using a Graph such as: https://www.cherrybiotech.com/scientific-note/accuracy-and-precision-in-measurements. Please clarify why it is crucial to understand these terms. I think it is basic knowledge for practitioners using 3D printing. Maybe you can underline the importance of these terms and give a connection between precision/trueness and possible error sources.
Lines 255-257: How did you measure the distortion? Can you quantify the distortion?
Lines 273-275: Explanations about the heatmap mode belong to M&M.
Line 299: Could you please explain the term quick-check GO NO-GO calibration gauges?
6.) Conclusion
Line 327ff.: Could you add some additional information about possible error sources linked with loss of trueness or loss of precision? This would give your manuscript a frame, as you introduced these terms. Just calling tech support is not a surprising solution. Maybe you can add some value to your manuscript by thoroughly interpreting your results.
Author Response
Thank you for taking the time to review our paper. We appreciate your valuable feedback. We have considered your comments and made the necessary changes to the manuscript.
1.) Title: As far as I know, "Geometric Dimensioning and Tolerancing" is a standardized system defined by the American Society of Mechanical Engineers, which does not only include metric measurements for defining the dimensional requirements for manufactured parts. GD&T describes nominal geometry and its allowable variation using a language or engineering drawings. The aim of GD&T is to add quality and reduce cost at the same time through producibility. It appears that the presented study uses a simpler way to assess the accuracy of printed objects and does not apply the rules stated in ASME Y14.5-2009. I think the methods presented have little connection to GD&T. Therefore, please consider revising the title.
The title has now been changed to ‘Monitoring the calibration of in-office 3D printers’.
2.) Abstract:
Line 22: Please specify that only two different resins were assessed: changes made to the manuscript and marked in red
3. Introduction:
Line 37: Please add references regardng the increasing attention in architecture, aeronautics and medicine.: changes made to the manuscript and marked in red
Lines 56-79: As this study investigates SLA printers it seems a bit off-topic to introduce all of them with multiple sentences. Moreover, after description of each printing methods you should add an appropriate reference, which enables the reader to proof your statements and gather further infomation (Only at some points in this paragraph Refernce 1 is indicated). I would suggest to drastically shorten this paragraph at line 55 and create a short informative table with the characterisitcs of each of the printing methods. Please add appropriate references in line 51-55 after first mentioning of each method:
changes made to the manuscript and marked in red
Line 80-85: Your statement is correct. Anyway in scientific literature you need to "proof" your statements by references. Please add current literature data: changes made to the manuscript and marked in red
Line 86-90: Please add reference regarding the shift to in-house printing: changes made to the manuscript and marked in red
Line 90-97: Very important and well written purpose of your study! :Thank you!
Line 103 ff. Please describe the term "worpiece with radii". Sentence was modified
Line 110-120: This paragraph contains the aim of your study (imprtant and correct!), but also some M&M descriptions (line 112-115). Please remove any aspects of this paragraph, which describe how you've performed your investigation. This belongs to M&M. Focus on describing your aims shortly and concise, without further justification. At the end of your introduction no further explanations should be needed to understand the study aims: Thank you for the suggestion. Changes made to the manuscript and marked in red
3.) M&M
2.1: please add an intrducing sentence before listing your material. Please specify the type of the used printer (Form 2?): changes made to the manuscript and marked in red
2.2 This paragraph (design of the coupon) should include all aspects that affected the design of the final coupon. Please move any descriptions about your preliminary experiences (mini rafts, exceeded tank etc.) into paragraph 2.2 and close with the description of the final coupon design: changes made to the manuscript and marked in red
Line 137: Add explanation and reference indicating that Z-axis is "not as critical" as the other axes: The sentence was re-worded to explain why Z axis was not included.
Figure 1: Consider Text editing. Captions should be at the same page as the figure. Please revise this figure. E.g.: mm (millimeter) and m (meter); spatial relations of the figure (a 6mm ring would appear much more slighter than presented, add the distances 16mm and 4mm --> where can they be measured?). It would be nice to have an appropriately captured screenshot of the meshmixer view instead of the "sketch" presented : New picture added with marked dimensions.
2.3: Consider deleting this paragraph after implementing all crucial aspects in 2.2! changes made to the manuscript and marked in red
Figure 2: Please specify what went wrong with each of the coupons. Upload higher resolution image please.
Figure 3: Upload higher resolution image please.
Figure 4: Again, resolution is too low. It would be helpful and nice for visualization to have one proper view ot the coupon as displayed in your design software and adding all explanatory captures. Why did you add numbers 1-4?
Figure 6/7: Figures are of low quality. Could you take a photo in a light box? Please correct captions of figure 6 (typo).
This study was done while I was still a resident at the university. I have now graduated and my mentor left the university as well. I no longer have access to the coupons we printed during the study and unfortunately, I am unable to click new photos to replace the ones I have added to the manuscript.
2.4
Lines 169-174: Consider splitting up this really long sentence. Btw do you mean "[...], as in corresponing studies.": changes made to the manuscript and marked in red
Lines 179-191: It does not besome clear which measurements were performed indetail. What have you done with the caliper? What was documented --> Please report in results! Is there any explanation why you frequently speak about curvature measurements and warping but do not measure it? If possible it would be helpful to have a statement/ quantification about the warping of the parts! You do mention that warping is crucial, this obligates you somehow to quantify/ describe it. Can you add this asprect?
- I have added a more detailed description of the caliper measurement and even added a picture to demonstrate the positioning of the jaws.
- I agree that it would be best to quantify the warping/deformation that we noticed. However, we did not include that in our measurements or analysis. I have re-worded my sentences regarding the warpage to give more clarity.
Line 189-191: Consider rewording or splitting up sentence: changes made to the manuscript and marked in red
Line 189: Fig. 6/6 seem not appropriate for visualizing the performed measurements. It would be helpful to have a figure describing the performed measurements: New image added to describe the positioning of the caliper. However, since I do not have access to the coupons anymore, I will be unable to click new photos.
) Results:
3.1 Unfortunately you do mention the importance of warpage for accuracy of 3D printing, but don't measure it. Apparently, the statistical analyses regarding x/y axes do not concern warpage. As already stated I would recommend to add measurements/ a statement about the quantity/extent of warpage. Mere visual examination is a comparable weak method.
I agree. I have eliminated that sentence and have re-worded the warpage description in the discussion.
Table 1: Please add the name of the actual analysis performed (Levene's test?) The p-value gives the level of significance but is not considered to be an an analysis itself. --> Same for Table 2 and 3. Could you please describe which measurements where compared with each other? Usually you set one measurement (e.g. position 1) as reference and compare it to the other. How did you perform the measurement? In general: please present the actual results of the measurement performed (using the caliper) in a concise table. This is a crucial drawback of your manuscript - it does not contain any numbers regarding the measurement, only statistical analysis. Please revise.
I have added the measurements in a table and included the statistical analysis in a separate paragraph
) Discussion
Lines 230-238: Consider using shorter sentences when introducing the terms trueness and precision. Maybe you can write it as it is usual in definition phrases. Consider using a Graph such as: https://www.cherrybiotech.com/scientific-note/accuracy-and-precision-in-measurements. Please clarify why it is crucial to understand these terms. I think it is basic knowledge for practitioners using 3D printing. Maybe you can underline the importance of these terms and give a connection between precision/trueness and possible error sources: changes made to the manuscript and marked in red
Lines 255-257: How did you measure the distortion? Can you quantify the distortion?
I have re-worded this section
Lines 273-275: Explanations about the heatmap mode belong to M&M: changes made to the manuscript and marked in red
Line 299: Could you please explain the term quick-check GO NO-GO calibration gauges?
I have added a more detailed description of the calibration gauge and added a picture as well.
6.) Conclusion
Line 327ff.: Could you add some additional information about possible error sources linked with loss of trueness or loss of precision? This would give your manuscript a frame, as you introduced these terms. Just calling tech support is not a surprising solution. Maybe you can add some value to your manuscript by thoroughly interpreting your results: changes made to the manuscript and marked in red

Round 2
Reviewer 2 Report
Thank you for reviewing the manuscript "Monitoring the Calibration of In-Office 3D Printers". There are still aspects of the manuscript which have to be improved prior to publication. As already mentioned the article lacks detailed references about the statements made in the introduction and discussion. Instead of indicating the same references over and over it would be nicer to have more appropriate literature citated parenthetically. The number of references may exceed up to 30 in a well-written article. I appreciate your honesty regarding the low qualitiy of the picture, nonethelss it would be nice to "tune" them a bit (at least with captions inside the figure e.g.) to improve readability.
1) Please remove the Term "Geometric Dimensioning and Tolerancing (GD&T)" completely in the manuscript (see lines 18 and 86).
2) Line 33: Please cite original pubication of Charles Hull.
3) Is reference 3 also valid to proof the statement that STL received attention in aeronautics and architecture?
4) Line 54: Very concise and useful table. Thank you.
5) Line 63: "For example" --> "for example"
6) Line 85: Delete GD&T as term. Instead introduce the performed evaluation as follows: "This study aims to develop a calibration coupon to assess the effect of different resins used for printing and the position of the test object on the build platform on printing accuracy."
7) Line 108: Here again: curvature variances are mentioned but not assessed in this study. Please reference Fig. 1 after this sentence.
8) Delete Lines 111-116 ("The final size [...]" until "[...] This was then subjected to further tsting as mentioned below").
9) Line 123: Delete the sentence: "There were no noticeable deformations present, and the overall process was sucessful" as this is a result and contains interpretation.
10) Line 120/ Line 130: Please introduce the abbreviations AMIST and ULSD before using the abbreviation for the first time. Why did you perform the production of the coupons at different locations? Where there any differences?
11) Figure 2: Can you describe why the alterations/ damages of each of the coupons occured? Is this all due to the resin tank?
12) Lines 155-160: This would be more appropriate in section 2.2 "Designing the coupon", wouldn't it?
13) Fig. 4: The engravings can't be recognized on the given picture. Consider adding additional letters/ numbers with Powerpoint e.g.
14) Lines 179-184: Create a different section 2.4 "Measurements" to describe the conducted measurements.
15) Lines 184-188: Create a different section 2.5 "Statistical analyses" to describe the statistical analyses conducted. Please make sure that you describe all the analyses conducted in detail.
16) Line 196: As already mentioned: Make sure to not split up figures and their captions on different pages.
17) Results: Please start introducing your descriptive results with a few sentences before placing the table.
18) Line 201ff.: Which test was used to assess statistical significant differences (t-test?) --> Please mention in M&M 2.5
19) Lines 218-220: Which test was used to assess statistical significant differences (t-test?) --> Please mention in M&M 2.5
20) Lines 221-224: This is an interpretaion of the results and belongs to the discussion.
21) As already mentioned: Please indicate the used analysis. P-value measurements are not a proper description. The p-value is only a value indicating the significance of a certain statistical test. It still remains unclear how the comparison of the different locations on the build platform were conducted. E.g. a p-value of 0.197 for position 1 (table 4) means that there is no difference of the variance observed at position 1 compared to ... (mean variance??). --> Mention in M&M 2.5
22) Lines 243-249: Well written summary how to use the terms precision/ trueness and tolerance in the dental field. Thank you!
23) Line 259: Again: What do you mean with curvature --Y When you mean warp it's not investigated in your study.
24) Line 273-279: Add references regarding the introduction of a new resin with higher elastic modulus.
25) Lines 350-354: Very good!
Author Response
Thank you for reviewing the manuscript "Monitoring the Calibration of In-Office 3D Printers". There are still aspects of the manuscript which have to be improved prior to publication. As already mentioned the article lacks detailed references about the statements made in the introduction and discussion. Instead of indicating the same references over and over it would be nicer to have more appropriate literature citated parenthetically. The number of references may exceed up to 30 in a well-written article. I appreciate your honesty regarding the low qualitiy of the picture, nonethelss it would be nice to "tune" them a bit (at least with captions inside the figure e.g.) to improve readability.
- Thank you for taking the time to review our work. Your comments and feedback have been very helpful in improving the relevance and readability of the manuscript!
- More references have been added to the introduction and discussion sections. The total number is now 30.
1) Please remove the Term "Geometric Dimensioning and Tolerancing (GD&T)" completely in the manuscript (see lines 18 and 86): Done
2) Line 33: Please cite original pubication of Charles Hull: Done and marked in red
3) Is reference 3 also valid to proof the statement that STL received attention in aeronautics and architecture? : Appropriate reference has been added
4) Line 54: Very concise and useful table. Thank you. : Thank you!
5) Line 63: "For example" --> "for example": Done and marked in red
6) Line 85: Delete GD&T as term. Instead introduce the performed evaluation as follows: "This study aims to develop a calibration coupon to assess the effect of different resins used for printing and the position of the test object on the build platform on printing accuracy.": Thank you for the suggestion. Change marked in red.
7) Line 108: Here again: curvature variances are mentioned but not assessed in this study. Please reference Fig. 1 after this sentence. : Changes marked in red
8) Delete Lines 111-116 ("The final size [...]" until "[...] This was then subjected to further tsting as mentioned below"): This part was added based on the comment of ‘Reviewer 1’.
9) Line 123: Delete the sentence: "There were no noticeable deformations present, and the overall process was sucessful" as this is a result and contains interpretation.: Done
10) Line 120/ Line 130: Please introduce the abbreviations AMIST and ULSD before using the abbreviation for the first time. Why did you perform the production of the coupons at different locations? Where there any differences?:
- The coupons were first printed at AMIST on an Anycubic photon 3D printer to test the overall printability. Since they did not have a Form 2 3D printer at the time, the second round of experimental printing was done at ULSD using this printer. This was done because the final study was to be completed on the Form 2 printer. Changes are marked in red.
11) Figure 2: Can you describe why the alterations/ damages of each of the coupons occured? Is this all due to the resin tank? Changes marked in red
12) Lines 155-160: This would be more appropriate in section 2.2 "Designing the coupon", wouldn't it?
- Yes. It has now been added to section 2.2.
13) Fig. 4: The engravings can't be recognized on the given picture. Consider adding additional letters/ numbers with Powerpoint e.g.
- Letters and numbers have been added using powerpoint.
14) Lines 179-184: Create a different section 2.4 "Measurements" to describe the conducted measurements.
- Good idea. New section created for measurements.
15) Lines 184-188: Create a different section 2.5 "Statistical analyses" to describe the statistical analyses conducted. Please make sure that you describe all the analyses conducted in detail.: Changes marked in red
16) Line 196: As already mentioned: Make sure to not split up figures and their captions on different pages. Done
17) Results: Please start introducing your descriptive results with a few sentences before placing the table: Introduction added and marked in red.
18) Line 201ff.: Which test was used to assess statistical significant differences (t-test?) --> Please mention in M&M 2.5
- Levene's test was used. Marked in red
19) Lines 218-220: Which test was used to assess statistical significant differences (t-test?) --> Please mention in M&M 2.5
- Levene's test was used. Changes marked in red
20) Lines 221-224: This is an interpretaion of the results and belongs to the discussion: Moved to discussion
21) As already mentioned: Please indicate the used analysis. P-value measurements are not a proper description. The p-value is only a value indicating the significance of a certain statistical test. It still remains unclear how the comparison of the different locations on the build platform were conducted. E.g. a p-value of 0.197 for position 1 (table 4) means that there is no difference of the variance observed at position 1 compared to ... (mean variance??). --> Mention in M&M 2.5
- X-axis position 1 for Dental SG was compared to X-axis position 1 for Grey V4 and that is how the rest of the measurements were made for the rest of the positions. This was then followed by Y-axis measurements.
- This is now added to the statistical analysis section and marked in red.
22) Lines 243-249: Well written summary how to use the terms precision/ trueness and tolerance in the dental field. Thank you!: Thank you!
23) Line 259: Again: What do you mean with curvature --Y When you mean warp it's not investigated in your study:
- Omitted the curvature section
- Changed the description for ‘warp’
24) Line 273-279: Add references regarding the introduction of a new resin with higher elastic modulus.: Reference added
25) Lines 350-354: Very good!: Thank you!
Round 3
Reviewer 2 Report
Dear Authors of the article "Monitoring the Calibration of In-Office 3D Printers",
I appreciate your persistent work on the article and hope that you also think that the manuscript was already improved significantly in its scientific nature compared to its initial form. Nevertheless some minor changes should be peformed before publication:
1) Line 21f: Consider rewording of the sentence. Maybe: " [...] on the build platform to assess errors caused by different positioning."
2) Line 55ff.: Please format the table consistently with the style of tables 3-5.
3) Line 61: Please add https://doi.org/10.3390/app11146444 besides reference 21.
4) Line 110: Consider rewording these sentences. ;Aybe: "However, curvature variances were not assessed in this study (Figure 1), because Z-axis is considered to be controlled by the mechanical components of the build platform and is therefore not an issue of calibration."
5) Line 120f. Caption of figure 1: I think this is the "final" design of the coupon? Please correct if applicable. Could you please visualize the thickness of 2mm of the calibration coupon? You can easily adopt the figure using Powerpoint or an open source graphic program. The caption of fig. 1 states that the inner diameter is 6mm. In line 148 it was stated that the inner diameter is 4mm. Please correct one of the numbers.
6) Line 130: Please indicate figure 5 after the sentence in which you describe the design of mini rafts. Please change order and numbering of the figures as they should be indicated in the same order as they appear in the manuscript.
7) Line 153ff. Consider rewording these sentences. Maybe: "Numbers "1" to "4" were engraved on the circular part of the coupon. The numbers will allow for orientation after taking the coupons off the platform and enable to perform measurements in defined areas of the coupon."
8) Line 209: Fig. 8: Please add letters with powerpoint or another software as they are not recognizable when the manuscript is printed out.
9) Line 135. Table 2: Please format the table consistently with the style of tables 3-5. Please "compress" the table in length by reducing line pitch. Please specifiy the mean value of all 10 printing cycles for x/y acis for both resins in an additional line at the end of the table. This will enable for a quick comparison without looking at each single value.
10) Line 251 & 254: Please edit table 4 & 5. Second column should just say "p-value". Pleae edit the captions: "Significance of Levene's test when comparing different positions on the build platform at the X/Y axis given by the according p-value."
10) Line 295ff. Mention the resins by name to facilitate readibality. "[...] it was seen that the coupons printed with Dental SG (Formlabs; Somerville, MA, USA) were less true to the original dimension of the coupon (up to 0.5 mm discrepancy) compared to the Grey V4 resin (Formlabs; Somerville, MA, USA; up to 0.26 mm discrepancies; Table 2)."
11) Line 354ff. Delete the sentence: "Since this is the first calibration study of its kind [...]." as this is not the first study adressing this issue.
Author Response
Thank you again for your patience to review our manuscript. We can definitely appreciate the overall improvement in the scientific quality and readability of the manuscript post these changes.
1) Line 21f: Consider rewording of the sentence. Maybe: " [...] on the build platform to assess errors caused by different positioning." : Done
2) Line 55ff.: Please format the table consistently with the style of tables 3-5.: Done
3) Line 61: Please add https://doi.org/10.3390/app11146444 besides reference 21. Done
4) Line 110: Consider rewording these sentences. ;Aybe: "However, curvature variances were not assessed in this study (Figure 1), because Z-axis is considered to be controlled by the mechanical components of the build platform and is therefore not an issue of calibration." Changes marked in red
5) Line 120f. Caption of figure 1: I think this is the "final" design of the coupon? Please correct if applicable. Could you please visualize the thickness of 2mm of the calibration coupon? You can easily adopt the figure using Powerpoint or an open source graphic program. The caption of fig. 1 states that the inner diameter is 6mm. In line 148 it was stated that the inner diameter is 4mm. Please correct one of the numbers.: New figure added with correct dimensions
6) Line 130: Please indicate figure 5 after the sentence in which you describe the design of mini rafts. Please change order and numbering of the figures as they should be indicated in the same order as they appear in the manuscript. Done
7) Line 153ff. Consider rewording these sentences. Maybe: "Numbers "1" to "4" were engraved on the circular part of the coupon. The numbers will allow for orientation after taking the coupons off the platform and enable to perform measurements in defined areas of the coupon." : Changed marked in red
8) Line 209: Fig. 8: Please add letters with powerpoint or another software as they are not recognizable when the manuscript is printed out.: Done
9) Line 135. Table 2: Please format the table consistently with the style of tables 3-5. Please "compress" the table in length by reducing line pitch. Please specifiy the mean value of all 10 printing cycles for x/y acis for both resins in an additional line at the end of the table. This will enable for a quick comparison without looking at each single value.: Changes marked in red
10) Line 251 & 254: Please edit table 4 & 5. Second column should just say "p-value". Pleae edit the captions: "Significance of Levene's test when comparing different positions on the build platform at the X/Y axis given by the according p-value.": Changes marked in red
10) Line 295ff. Mention the resins by name to facilitate readibality. "[...] it was seen that the coupons printed with Dental SG (Formlabs; Somerville, MA, USA) were less true to the original dimension of the coupon (up to 0.5 mm discrepancy) compared to the Grey V4 resin (Formlabs; Somerville, MA, USA; up to 0.26 mm discrepancies; Table 2)." Changes marked in red
11) Line 354ff. Delete the sentence: "Since this is the first calibration study of its kind [...]." as this is not the first study adressing this issue.: Done